# Targeting Cellular Retinoic Acid Binding Protein 1 with Retinoic Acid-like Compounds to Mitigate Motor Neuron Degeneration

**DOI:** 10.3390/ijms24054980

**Published:** 2023-03-04

**Authors:** Jennifer Nhieu, Liming Milbauer, Thomas Lerdall, Fatimah Najjar, Chin-Wen Wei, Ryosuke Ishida, Yue Ma, Hiroyuki Kagechika, Li-Na Wei

**Affiliations:** 1Department of Pharmacology, University of Minnesota, Minneapolis, MN 55455, USA; 2Institute of Biomaterials and Bioengineering, Tokyo Medical and Dental University, 2-3-10 Kanda-Surugadai, Chiyoda-ku, Tokyo 101-0062, Japan

**Keywords:** CRABP1, retinoic acid, ligand, CaMKII, non-canonical, neurodegeneration, motor neuron

## Abstract

All-trans-retinoic Acid (atRA) is the principal active metabolite of Vitamin A, essential for various biological processes. The activities of atRA are mediated by nuclear RA receptors (RARs) to alter gene expression (canonical activities) or by cellular retinoic acid binding protein 1 (CRABP1) to rapidly (minutes) modulate cytosolic kinase signaling, including calcium calmodulin-activated kinase 2 (CaMKII) (non-canonical activities). Clinically, atRA-like compounds have been extensively studied for therapeutic applications; however, RAR-mediated toxicity severely hindered the progress. It is highly desirable to identify CRABP1-binding ligands that lack RAR activity. Studies of CRABP1 knockout (CKO) mice revealed CRABP1 to be a new therapeutic target, especially for motor neuron (MN) degenerative diseases where CaMKII signaling in MN is critical. This study reports a P19-MN differentiation system, enabling studies of CRABP1 ligands in various stages of MN differentiation, and identifies a new CRABP1-binding ligand C32. Using the P19-MN differentiation system, the study establishes C32 and previously reported C4 as CRABP1 ligands that can modulate CaMKII activation in the P19-MN differentiation process. Further, in committed MN cells, elevating CRABP1 reduces excitotoxicity-triggered MN death, supporting a protective role for CRABP1 signaling in MN survival. C32 and C4 CRABP1 ligands were also protective against excitotoxicity-triggered MN death. The results provide insight into the potential of signaling pathway-selective, CRABP1-binding, atRA-like ligands in mitigating MN degenerative diseases.

## 1. Introduction

All*-trans*-retinoic acid (atRA) is the principal active metabolite of vitamin A with well-known biological activities in development, differentiation, apoptosis, and many other biological processes [1]. These activities of atRA are known to be primarily mediated by nuclear RA receptors (RARs) that act as transcriptional factors to alter gene expression [1,2] and are referred to as atRA’s canonical activities. Recently, it has been shown that atRA also elicits RAR-independent activities that can be detected rapidly (within minutes) in the cytoplasm [2,3,4], referred to as non-canonical activities [3,4]. Using a gene knockout approach, it has been established that the non-canonical activities of atRA are mediated by a specific high-affinity cytosolic atRA-binding protein named cellular retinoic acid binding protein 1 (CRABP1) [5,6]. 

CRABP1 is a highly conserved cytosolic protein and is expressed in multiple cell types, including embryonic stem cells (ESCs) [5], cardiomyocytes [7], adipocytes [8,9], and motor neurons (MNs) [10], etc. In ESC, CRABP1-RA modulates cell cycle progression, and deleting CRABP1 from ESC accelerates cell cycle progression [5], supporting the notion that CRABP1 can be a tumor suppressor [11,12]. In cardiomyocytes, CRABP1 protects cardiomyocytes from apoptosis triggered by adrenergic over-stimulation; therefore, *Crabp*1 knockout (CKO) mice are prone to isoproterenol-induced cardiomyopathy and heart failure [7]. In adipocytes, CRABP1 facilitates adiponectin secretion and mitochondria homeostasis; therefore, CKO mice have significantly reduced adiponectin levels and are prone to high fat diet-induced adipose inflammation [9]. In MNs, CRABP1 protects against neuronal stress/death, and CKO mice spontaneously develop adult-onset progressive motor deterioration, mimicking amyotrophic lateral sclerosis (ALS) due to progressive MN death and neuromuscular junction defects [10]. While CRABP1-RA appears to affect various cell types and deleting CRABP1 causes different pathological outcomes in various organ systems, it is interesting that pathologies caused by CRABP1 deletion are related to two conserved signaling pathways, i.e., extracellular signal-regulated kinase (Erk) and calcium (Ca^2+^) calmodulin-activated kinase 2 (CaMKII). 

In Erk kinase activation, CRABP1 directly interacts with rapidly accelerated fibrosarcoma 1 (Raf-1), which is the first kinase component in the mitogen-activated protein kinase (MAPK) signaling pathway, to ultimately dampen mitogen or growth factor-stimulated Erk activation. This is crucial to numerous cellular processes, especially growth [11]. CRABP1 modulation of Erk signaling plays out in the context of stem cell proliferation (in ESCs and tumors) [5,12], and for adiponectin secretion (in adipocytes) [9]. In CaMKII activation, CRABP1 directly interacts with CaMKII to dampen its enzyme activation, thereby preventing over-stimulation of CaMKII in excitable cells (such as cardiomyocytes and MNs) and protecting them from cytotoxicity and cell death [7,10]. This is supported by the finding that the physiological ligand of CRABP1, atRA, can be used to protect against isoproterenol-induced cardiomyopathy in wild-type mice but not in CKO mice, and suggests a therapeutic potential of atRA in specifically targeting CRABP1 to reduce CaMKII over-activation related pathologies [13]. However, given the well-known toxicity of atRA [14,15], via RAR activation, in long-term applications, it is desirable to explore atRA-like compounds that can specifically target CRABP1 without activating RARs in order to avoid retinoid toxicity. 

To further validate that CRABP1 can be a useful therapeutic target in managing pathological conditions caused by CaMKII over-activation, we recently carefully examined the ALS-like motor deterioration phenotype of CKO mice and the mechanism of CRABP1 action in the motor system [10]. It appears that CRABP1 is specifically expressed in spinal MNs, and elevating its expression in MNs to dampen CaMKII activation is beneficial to the neuromuscular junction (NMJ) health, partially attributable to enhanced MN agrin expression and axon extension. Specifically, CaMKII is aberrantly activated in the spinal MN population of adult CKO mice, and re-introducing CRABP1 to young CKO mice could significantly lower their CaMKII activity in spinal MNs and rescue their motor defects later. In a preliminary in vitro test using an immortalized MN cell line, MN1, we found that elevating CRABP1 levels in MN1 improved their axon extension. These experiments provided further evidence for the potential therapeutic application of targeting CRABP1, such as in dealing with diseases caused by defects in MNs [10]. Extended from these interesting findings in the CKO mouse model, this current study aims to search for CRABP1-binding (without activating RAR), atRA-like compounds that can modulate CaMKII and to establish an in vitro model for studying if and how these CRABP1-ligands may affect the process of MN differentiation and health. 

While the literature has reported several MN model systems [16], there remains a need for a more reliable and reproducible in vitro system where MN differentiation can be more robustly induced for systemic studies, especially studies allowing dissection of intermediate events in various stages of MN differentiation and maturation processes. For this current study, we, therefore, exploited P19, a widely used embryonal carcinoma cell line that is embryonic stem cell (ESC)-like and much easier to manipulate; further, P19 does not require a feeder layer in culture [17]. Using this newly developed P19-MN differentiation system, we tested the feasibility of targeting CRABP1 to modulate CaMKII activation in the process of MN differentiation and in maintaining MN health.

## 2. Results

### 2.1. Characterization of C32 as Novel atRA-like Compound That Binds CRABP1

Previously, using a rational screening approach, we have identified novel, atRA-like compounds, C3 and C4, as CRABP1 ligands that modulate Erk activation [12]. Using this approach, we define a hit CRABP1-binding compound as (1) binding to CRABP1 and (2) lacking RAR activation activity. We now report another CRAPB1 ligand, C32 (chemical name: 2-(3,5,5,8,8-Pentamethyl-5,6,7,8-tetrahydronaphthalene-2-carboxamido) thiazole-5-carboxylic acid). The chemical structures of atRA, C32, and C4 are shown in Figure 1A. Differential scanning fluorimetry (DSF) was utilized to determine CRABP1-binding activity. DSF is a thermal shift assay in which ligand binding to the protein of interest results in a thermal-stable, ligand-protein complex with a higher melting temperature compared to that of vehicle control [18]. The relative increase in melting temperature upon ligand binding is reported as delta Tm (ΔT_m_). In this assay, CRABP1 binding was defined as a ΔT_m_ greater than or equal to 1 °C (ΔT_m_ ≥ 1 °C). Typical high-throughput screening approaches utilize a cut-off of 3 standard deviations above the mean [19,20,21]. Given that CRABP1 has a highly consistent Tm (Appendix A), with minimal variance across biological and technical replicates, ΔTm ≥ 1 °C provides a highly stringent cut-off for the hits. This high stringency cut-off allows the identification of robust CRABP1-binding compounds and reduces the potential for false positives.

This criterium for CRABP1 binding is described in-depth in Section 4.2. In DSF, atRA (100 μM) was first used as a positive control (Figure 1B). When compared to the DMSO control (blue curve), RA was detected to bind CRABP1 (red curve), generating a ΔT_m_ = 20 °C, validating this DSF assay in detecting CRABP1-binding compounds. Using this test, C32 (100 μM) appeared to bind CRABP1, generating a ΔT_m_ = 3 °C (orange curve) compared to the DMSO control (blue curve) (Figure 1C). Previously identified, using a conventional ligand displacement assay [12], CRABP1 ligands C3 and C4 were also subjected to DSF (Appendix A). C3 and C4 at 100uM generated ΔT_m_ = 0.26 and ΔT_m_ = 0.09, respectively. Although these values are below the CRABP1 binding criterium of ΔT_m_ ≥ 1 °C, the positive shift in Tm indicates a CRABP1-binding event for both C3 and C4, consistent with the previous positive result using conventional ligand displacement assay [12]. These results suggest that DSF with a stringent cut-off of ΔTm ≥ 1 °C is suitable for identifying ligands that robustly bind CRABP1. 

To ensure that C32 met the second criterion, i.e., lacking RAR-activation activity, we utilized a classical luciferase-based RAR activation assay in the Cos-1 cell line. As expected, RA (0.25 μM) significantly activated RAR (298 ± 83.5-fold activation) compared to the control, whereas C32 exhibited no RAR activation (0.76 ± 0.47-fold-activation) compared to the control (Figure 1D). These data allowed us to identify C32 as a new atRA-like compound that binds CRABP1 without activating RAR.

### 2.2. AtRA and C32 in CRABB1-Mediated CaMKII Dampening

CaMKII activity is marked by changes in the phosphorylation status of key regulatory residue threonine 286 (or 287, depending on the isoform) [22]. The regulatory role for CRABP1 in CaMKII activation was first examined in a reconstituted HEK293T cell model transfected with CaMKII and CRABP1 (or empty vector control). The status of pCaMKII was then assessed via western blot using an antibody specific for phosphorylated threonine 286/7. Using this system, we then determined the CaMKII-modulating activities of the three CRABP1-binding ligands, C32 and two previously reported ligands, C3 and C4 that were shown to elicit CRABP1-dependent Erk-modulating activity. The results show that atRA, C32, and C4 treatment had no significant effect on pCaMKII activity in the control vector transfected cells (Figure 2A, left blot; Figure 2B left graph). However, treatment with atRA, C32, or C4 at 0.5–5 μM for 15 min dampened CaMKII activity in CRABP1-transfected cells (Figure 2A, right blot; Figure 2B right graph), whereas C3 (that could elicit CRABP1-mediated Erk modulation) had no effect on CaMKII activation. Therefore, C32 and C4, as well as the positive control atRA, are CRABP1 ligands that can dampen CaMKII activity. This is further supported by the fact that this CRABP1-dependent CaMKII dampening activity is detected rapidly (15–60 min), confirming that this CRABP1-dependent C32 and C4 activity is non-canonical in nature. 

We then sought to determine if this non-canonical activity of atRA, C32, and C4 on CaMKII modulation can be detected in a more biologically relevant context in which CRABP1 and CaMKII are endogenously present. We thus exploited the widely used P19 embryonal carcinoma cell line, which was shown to endogenously express both CRABP1 [23] and CaMKII [24]. In contrast to the reconstituted HEK293T cell studies where only a single isoform CaMKII isoform (CaMKII beta) was introduced, P19 cells endogenously expressed two isoforms of CaMKII, indicated by the detection of two bands with the pan-pCaMKII antibody. Four major isoforms of CaMKII -alpha, beta, delta, and gamma are known to exist and vary in expression levels depending on cell and tissue types [25], stage of development [26], and other biological and disease contexts [27,28]. 

Treatment with atRA, C32, or C4 at 0.5–5 μM for 15 min dampened endogenous CaMKII in P19 cells (Figure 2C,D), validating this CaMKII-modulatory effect of CRABP1 ligands in a physiological context. Therefore, it is concluded that C32 and C4 elicit CRABP1-dependent CaMKII-modulatory (dampening) activity. 

### 2.3. A New In Vitro Stem Cell-MN Differentiation System, P19, for Studying CaMKII Activation

We have previously reported that CKO mice exhibited dramatically elevated CaMKII activation in MNs, which contributed to MN death and motor deterioration in adult mice [10]. This prompted us to carefully examine how CRABP1 signaling might affect the MN differentiation process. We thus exploited the P19 cell line, which has been extensively utilized for its differentiation potential in studying various stages of cell differentiation, including neuronal differentiation [17]. Here we developed a P19-derived motor neuron (MN) differentiation culture system to probe the effects of atRA and CRABP1 ligands in MN differentiation with regard to CaMKII activation. This P19 culture system is superior to ESC-differentiation systems which generally are very sensitive to technical complications and clonal variation [16,29]. 

Figure 3A depicts the workflow for the new P19-MN differentiation system and the data demonstrating MN differentiation efficiency. First, P19 cells in a single-cell suspension were transferred into P19 Differentiation Medium containing 0.5 μM atRA (+RA Medium, see Section 4.6 for complete media formulations) in a T75 flask. The flask was stored up-right to prevent cells from attaching to the coated surface of the flask, allowing the cells to form embryoid bodies (EB) over a two-day period (Day −4 to Day −2). The EBs were then collected and resuspended in fresh P19 Differentiation Medium containing 0.5 μM atRA and 200 ng/mL mouse Sonic Hedgehog protein (Shh) (+RA, +Shh Medium) in a new T-75 flask for neurosphere (NS) formation over the next two-day period (Day −2 to Day 0). The flask was stored in the same manner to promote NS formation. Shh is a potent morphogen known to promote ESC differentiation into functional MNs [30]. On Day 0, the NSs were dissociated into single cells and then plated onto a Matrigel-coated 6-well plate for MN differentiation (Day 1 to Day 3). Cells were collected and analyzed at various time points for the expression of relevant markers of MNs. 

Figure 3B shows brightfield images of undifferentiated P19 stem cells (Day −4, left) and P19-derived Day 3 MNs (right). Undifferentiated P19 cells typically grow in a clumped manner with an epithelial-like morphology. In contrast, Day 3 P19-MNs grow elongated and branched processes. In addition to MN-like morphological features, the induction of several MN-specific marker genes, *HB9* [31], *ChAT* [32], *Isl1*, and *Isl2* [33,34] (Figure 3C, Appendix A) and markers for other spinal neurons such as V2 interneurons, *LHx3* [35] (Appendix A) were also monitored. The induction of *Isl2* specifically marks the presence of somatic-type spinal MNs, which innervate and maintain muscle tissue tone [36]. MN markers and spinal neuron markers appear to be readily elevated on Day 1 (Figure 3C, Appendix A), begin to decline after Day 3, and continue to decline on Day 5 (Appendix A). Peak expression of MN markers on Day 1–3 suggests that this could be the optimal time window to specifically study the effect of CRABP1-ligands in the MN-differentiation process with regards to CaMKII activation. 

Importantly, *Crabp1* gene expression appears to be elevated from the EB stage and steadily maintained until Day 3 (Figure 3D, Appendix A). On Day 5 of MN differentiation, a sharp decrease in *Crabp1* expression was observed (Appendix A), further supporting that Day 1–3 is the optimal time window to study the effects of CRABP1 ligands on CaMKII activation in this P19-MN differentiation system. 

### 2.4. AtRA, C32, C4 Dampening CaMKII Activity in P19-MN Differentiation Process

To determine if atRA, C32, or C4 affects CRABP1-mediated CaMKII modulation, we carried out a series of experiments focusing on Day 1 and Day 3 in the P19-MN differentiation system. First, on the relevant day of interest (Day 1 or Day 3), atRA and Shh were depleted by replacing the medium with fresh differentiation medium without atRA and Shh. Additionally, dextran charcoal-treated bovine serum was used to further deplete other factors, particularly retinoids, to remove any potential contribution of genomic activities of atRA. Following this depletion step, cells were treated with atRA, C32, or C4 and then immediately harvested for western blot analyses to monitor CaMKII activation (Figure 3A, open circles).

On Day 1 (Figure 4A,B) and Day 3 P19-MN (Figure 4C,D), atRA, C32, and C4 significantly dampened endogenous CaMKII activity. For Day 1 MNs, RA, C32, or C4 were added at 1 μM for 15 min. For Day 3 MNs, atRA, C32, or C4 were added at 1 μM for 30 min. Together, the data show that atRA and CRABP1-binding ligands, C32 and C4, can dampen CaMKII in both Day 1 and Day 3 P19-MN differentiating/differentiated cells, suggesting their effects in multiple stages of the MN differentiation process. Note that this CaMKII-dampening activity was also detected rapidly (15–60 min), confirming that they elicited the non-canonical activity of atRA. 

Interestingly, additional isoforms of CaMKII are routinely detected in P19-MNs, compared to undifferentiated P19 cells. This suggests expression of additional, neuron-specific alpha or beta CaMKII isoforms in more differentiated cells [25]. While the intensity of isoforms differed between D1 and D3 cells, all CaMKII isoforms were dampened by these CRABP1 ligands in both D1 and D3 cells, suggesting that CRABP-modulation is effective for these various CaMKII isoforms. 

### 2.5. CRABP1 in Neuroprotection against Calcium (Ca^2+^)-Induced Toxicity in MN Cells

Aberrant CaMKII activity is frequently implicated in neurodegeneration, especially in mediating the destructive downstream events of excitotoxicity, such as calcium (Ca^2+^) overload-mediated cell death [37]. We previously reported that MN cells in CKO mice had highly elevated CaMKII activity and appeared unhealthy, which also coincided with their augmented expression of agrin protein, a proteoglycan essential for MN development and health [10,38]. This suggests that CRABP1 could be a protective player in maintaining healthy MNs.

To determine if CRABP1 indeed plays a neuroprotective role in committed MN cells, such as during excitotoxic insult, we exploited an established MN-committed cell line (MN1). We generated a stable CRABP1-overexpressing MN1 clone (CRABP1-MN1) and determined if this could provide a protective effect against ionomycin assault. Ionomycin is a selective Ca^2+^ ionophore that causes rapid increases in intracellular Ca^2+^ concentration ([Ca^2+^]_i_) [39], triggering CaMKII activation [22], an event contributing to toxicity and subsequent cell death [40]. Wild-type (WT) MN1 and CRABP1-MN1 were exposed to ionomycin at 5 μM for 18 h, and cell viability was monitored immediately after the 18 h incubation period with MTT using 3-(4,5-dimethylthiazol-2-yl)-2,5-diphenyltetrazolium bromide (MTT) viability assay. MTT is metabolized by living cells to an insoluble, measurable form known as formazan [41], which directly measures live cell/viability. Indeed the results show that CRABP1-MN1 exhibits significantly greater cell viability, as compared to WT, after ionomycin exposure (Figure 5A), confirming that elevating CRABP1 levels can protect MN from excitotoxicity-induced cell death.

To validate the neuroprotective, biological activity of CRABP1 ligands, atRA, C32, and C4, we determined if these ligands could protect against ionomycin assault in WT MN1 cells, a more physiologically relevant experimental system. Because this experiment is to determine the potential protective effects of the compounds, endogenous levels of CRABP1 expression provide a more biologically relevant cell context to study the effects of these compounds. First, MN1 cells were pre-treated with atRA, C32, or C4 (0.5–5 μM) for 1.5 h. Immediately after pre-treatment, ionomycin (4 μM) or DMSO (as vehicle control) was added, and cells were incubated overnight, which typically induced cell death. Compounds were present during the duration of ionomycin exposure. The next day, treated cells were subjected to an MTT assay. As expected, compared to DMSO control, ionomycin (4 μM) significantly reduced cell viability. Interestingly, atRA exhibited a trend towards improved cell viability, while C32 and C4 significantly improved cell viability (Figure 5B).

To demonstrate the proposed mechanism via CRABP1-mediated CaMKII dampening, we compared Wild-type (WT) and CRABP1-MN1 (over-expressing CRABP1 to elevate CRABP1 level) treated with medium (basal), DMSO (vehicle control), or ionomycin (10 μM, 5–10 min) with regards to their CaMKII activation as reflected on threonine 286/7 phospho-status. As shown in Figure 5C, elevating the CRABP1 level (right panel, CRABP1 Over-Expression) clearly dampened CaMKII activity. This result supports the observations made previously using the reconstituted HEK293T system [10]. This further strengthens our hypothesis that targeting CRABP1-signaling can be developed into a protective and/or therapeutic strategy.

Taking data collected from P19-MN differentiation and MN1 systems, it is concluded CRABP1 can provide a neuroprotective effect against cell death induced by pathological Ca^2+^ overload. Furthermore, this protective effect can be exploited by using CRABP1 ligands such as C32 and C4 to enhance the protective mechanism and improve cell viability. Mechanistically this neuroprotective effect can be attributed to CRABP1-signaling that dampens CaMKII over-activation in differentiating or differentiated MNs. In summary, as depicted in the proposed model (Figure 5D), when MNs experience cytotoxic stimulation (step 1), it can result in pathological increases in [Ca^2+^]_i_ and subsequent CaMKII activation and phosphorylation of AMPAR [42] (step 2), CRABP1, as well as its ligand (such as atRA, C32, and C4), could provide an inhibitory effect to dampen this aberrant CaMKII activation (step 3), thereby preventing MN death (step 4).

## 3. Discussion

Here we report the screening of new atRA-like compounds for binding to CRABP1 without activating RAR, which allowed us to identify a new CRABP1 ligand, C32. By testing C32, as well as previously reported CRABP1-binding compounds [12] for their ability to modulate CaMKII activation, we have identified C32 and C4 (previously reported to modulate Erk activation) [12], both are CRABP1-binding ligands that can modulate CaMKII signaling in MNs. In our previous reports, C3 and C4 were identified based on their binding to CRABP1, detected using conventional ligand displacement assay, and their ability to modulate Erk signaling [12]. Together, these results show that C32 exhibits CaMKII-modulatory activity, C3 is an Erk-selective CRABP1 ligand, whereas C4 appears to be a pan-acting CRABP1 ligand. How C32 behaves in Erk signaling will require further intensive study. Nevertheless, these three compounds comprise the first series of useful CRABP1-binding ligands that may be worthy of further investigation. In order to describe the precise binding profiles of C32, C3, C4, and next-generation CRABP1-binding compounds, future studies are needed to rigorously compare their pharmacological properties. For instance, it would be of most interest to determine their binding characteristics, such as affinity and kinetics, in order to more precisely differentiate and categorize CRABP1 hit compounds. These studies will also be important for generating more rationale-designed, next-generation CRABP1 compounds that could display pathway-selective compounds. To this end, assays to confirm the various biological activities of CRABP1 ligands will be important in order to contextualize the therapeutic potential and physiological relevance of these novel CRABP1 ligands. An important feature of these compounds is their lack of RAR-activation ability (see the following section). Preclinical models would be important for further studies to determine their potential as therapeutics by targeting CRABP1 to mitigate diseases associated with the over-activation of Erk, CaMKII, or both signaling pathways. We previously hypothesized that it is possible to design CRABP1-specific and signaling pathway-selective atRA-like compounds for safer therapeutic applications [5,12]. This current study supports such an interesting possibility.

AtRA and atRA-like compounds targeting RARs have been extensively and enthusiastically studied for therapeutic applications; however, the widely documented toxicity (RAR-mediated retinoid toxicity) has hindered the progress in this field and greatly limited their potential in clinical applications. The recently established non-canonical activities of atRA, mediated by CRABP1 [43], and the demonstration of multiple human disease-mimicking phenotypes of CKO mice [7,10] prompted us to propose a new therapeutic strategy using CRABP1-binding, atRA-like compounds that lack RAR activity to modulate specific disease-related signaling pathways such as Erk and CaMKII. These compounds were designed based on, specifically, the CRABP1 binding pocket, and it is known that the binding pocket of CRABP1 has little structural or sequence relationship with the ligand binding domain of RAR [44,45,46,47]. According to the original design strategy, it is tempting to speculate that C3, C4, and C32 may not bind RAR. However, for truly “CRABP1-specific” ligands, RAR binding and potential antagonism must be ruled-out in future studies. Nevertheless, this current study provides the first support for the possibility of designing signaling pathway-selective CRABP1-binding ligands for therapeutic intervention.

With regards to CRABP1 modulating CaMKII signaling, we have reported two spontaneously developed human disease-mimicking phenotypes of CKO mice; both are associated with aberrant CaMKII activation and cytotoxicity, i.e., cardiomyopathy/heart failure caused by cardiomyocyte apoptosis and death [7] and ALS-like motor deterioration caused by MN death/loss [10]. Importantly, in human studies, drastically reduced *Crabp1* gene expression has been reported in neurodegenerative disease patients, including ALS and SMA patients [48,49]; therefore, we prioritized the studies of CRABP1-binding ligands in the context of MN degeneration. Extensive classical studies have reported aberrant CaMKII activation in neurodegeneration because over-activation of CaMKII and abruptly surged intracellular Ca^2+^ concentration is often a result of excitotoxicity, which subsequently leads to neuron death [50]. As introduced earlier, in this current study, we extended our previous findings of CKO mouse MN degenerative phenotype, aiming to develop a more reliable and feasible in vitro MN model for mechanistic studies and for screening CRABP1-specific compounds that can modulate CaMKII to improve MN health. To this end, we were able to exploit P19 and develop a P19-MN differentiation system, which allowed us to identify C32 and C4 as CRABP1 ligands that could modulate CaMKII in the context of MN differentiation. Our data also support the notion that increasing CRABP1 signaling can improve MN health, as demonstrated by the reduction of excitotoxic-induced death in an MN1 cell line with stably elevated CRABP1 expression. We also show that C32 and C4 are protective against ionomycin assault (mimicking Ca^2+^ overload as observed in excitotoxicity). Mechanistically, this protection may be attributed to their ability to bind CRABP1 to further dampen CaMKII over-activation.

The potential role of CRABP1 in modulating MN health and motor function was first detected in CKO mice; however, it was unclear when CRABP1 and its ligands could play a role along the course of MN differentiation or maturation. Because MN1 is a committed MN cell line [51], this system is not appropriate for dissecting intermediate events during the differentiation, especially in the early differentiation stages. On the contrary, the P19-MN differentiation system spans the entire course of neural progenitor (or stem cell) progressing to MN differentiation, allowing interrogation of intermediate steps in the entire process of MN differentiation. As shown here, both C32 and C4 are effective, in terms of modulating endogenous CaMKII activation, at both early and late differentiation stages, suggesting that CRABP1 signaling can be involved in multiple steps of MN differentiation. Targeting CRABP1 may be beneficial to multiple steps in the MN differentiation process; therefore, this strategy may also be useful in preventive applications. Along with our findings that C32 and C4 pre-treatment improves cell viability after ionomycin assault in MN1 cells, it is tempting to speculate a potential clinical application as a prophylactic therapeutic to combating various neurological pathologies associated with Ca^2+^ overload/excitotoxicity. Preventative approaches may also offer an attractive strategy that can preserve a healthy neuronal population, in contrast to interventions that are typically introduced at disease onset when significant neuronal damage has already occurred [52].

Since both CaMKII and Erk signaling pathways are important for normal physiological processes in numerous organ systems and multiple cell types, pharmacological intervention to target these signaling pathways non-discriminatorily is more likely to cause toxicity. However, by selectively targeting one of these signaling pathways modulated by CRABP1 with CRABP1-specific ligands, it is possible to deliver therapeutic effects that are more specific and safer because these drug effects would be limited to certain disease-relevant cell types that are CRABP1-positive. Therefore, it would be interesting in future studies to address whether compounds like C32 and C4 can be used in therapeutic/preventive applications for diseases associated with CaMKII over-activation in CRABP1-positive cells/tissues, such as progressive motor deterioration and cardiomyopathy as revealed in the CKO mouse model. A pan-acting ligand like C4 may be useful in dealing with disease conditions where both Erk and CaMKII signaling pathways are altered, such as Alzheimer’s disease (AD), amyotrophic lateral sclerosis (ALS), and Parkinson’s disease (PD) [53,54,55,56].

The list of pathological conditions associated with the non-canonical activities of atRA is growing. This study provides the first insight into the potential of designing signaling pathway-selective, CRABP1-binding atRA-like ligands in mitigating diseases. In the future, CRABP1-specific compounds that can elicit the non-canonical activities of atRA may constitute an attractive and novel group of compounds that have the potential for therapeutic/preventive application for a wide spectrum of diseases with minimized retinoid toxicities.

## 4. Materials and Methods

### 4.1. Reagents and Compound Library

All-trans retinoic acid (atRA, Sigma Cat# R2625) and ionomycin salt (Cat# I0634) were obtained from Sigma (St. Louis, MO, USA) and dissolved in DMSO. RA-like compounds C32 and C4 were synthesized by our collaborator Dr. Hiroyuki Kagechika at Tokyo Medical and Dental University. More details on the RA-like compound library can be found in [12]. For compound studies, C32 and C4 were dissolved in DMSO. All compounds were stored at −80 °C with limited freeze-thaw cycles.

### 4.2. Differential Scanning Fluorimetry (DSF) CRABP1 Binding Assay and Data Analysis

His-tagged CRABP1 was purified as described in [11]. 5 ug of CRABP1 (14.1 μM) was incubated with 100 μM of atRA, C32, C3, or C4 to yield a ligand to CRABP1 molar ratio of 7:1 in order to achieve saturation and preserve ligand solubility in the aqueous reaction buffer. For all experiments, DMSO was used as the vehicle control, and binding reactions were carried out in 1XPBS, pH 8.0 buffer for 1 h at room temperature on an orbital shaker. After incubation, 18 μL of the CRABP1-ligand mixture was transferred into a 96-well plate and 2 μL of 20X SYPRO Orange was added to a final concentration of 2× SYPRO Orange and a final reaction volume of 20 μL in a single well. SYPRO Orange (Invitrogen Cat. S6650, Waltham, MA, USA) was diluted from a 5000xX stock to a 20× stock in 1XPBS, pH 8.0.

Data were acquired on the QuantStudio™ 3 Real-Time PCR instrument (Applied Biosystems, Waltham, MA, USA). Data acquisition parameters were created using Design and Analysis Software (Ver 2.6.0) and are as follows: (1) starting temperature of 25 °C held for 2 min, (2) temperature was then incrementally increased from 25 to 99 °C at a ramp speed of 0.05 °C/s, (3) fluorescent readings were acquired using filter settings of 520 ± 10 nm excitation wavelength and 558 ± 11 nm emission wavelength. For each independent experiment, 6 technical replicates were included for each condition. Wells containing only ligand and SYPRO Orange were assayed as “ligand-only” controls to ensure that compounds alone did not contribute any fluorescent signal that may interfere with data analysis or generate false positives.

For data analysis, CRABP1 melt curves were generated by calculating the negative first derivative of fluorescence (RFU) over temperature (T) (−ΔRFU/ΔT) and plotting against temperature. −ΔRFU/ΔT and temperature values were calculated in the Design and Analysis software. This melt curve data was then exported to Microsoft Excel, which was used to extract the minimum (lowest) −ΔRFU/ΔT value of the melt curve. The corresponding temperature to the minimum −ΔRFU/ΔT value defines the CRABP1 melting temperature (Tm) [57]. The Tm’s from the 6 technical replicates were averaged and used to calculate the thermal shift values (ΔTm). ΔTm was calculated by taking the difference between DMSO control and ligand conditions (ΔTm= ligand-DMSO). A hit compound was defined as a ΔTm greater than or equal to 1 °C above the DMSO control Tm (ΔTm ≥ 1 °C of DMSO). This cut-off was selected as a more stringent hit threshold to identify hit ligands that generated a robust ΔTm. Experiments were performed three independent times.

### 4.3. Cell Culture

Cos-1 cells were maintained as described in [12]. HEK293T cells were maintained in DMEM medium (Gibco #11965, Billings, MT, USA) containing 4.5 g/L D-glucose, 4 mM L-glutamine, 44 mM Sodium Bicarbonate, 100 U/mL penicillin, 100 mg/mL streptomycin, and 10% heat-inactivated FBS as described in [10]. HEK293T cells were co-transfected with GFP-CaMKII (Addgene #21227, Watertown, MA, USA) and either empty vector (EV) or Flag-CRABP1 (construct information described in [11]) and in a GFP-CaMKII to EV/CRABP1 to a ratio of 1:5. A total of 10 ug for a 10 cm cell culture dish was transfected using polyethylenimine (PEI, Polysciences Cat #23966, Warrington, PA, USA) in a DNA to PEI ratio of 1:5. P19 cells were purchased from ATCC, Manassas VA, USA (Cat# CRL-1825) and maintained in alpha minimum essential medium (MEM) with ribonucleosides and deoxyribonucleosides (Gibco, Cat# 12571-063) supplemented with 7.5% Bovine Calf Serum, Ion Fortified (ATCC, Cat # 30-2030), 2.5% Fetal bovine serum (R&D Systems, Cat# S11150, Minneapolis, MN, USA) and Pen Strep (Gibco, Cat #15140-122). All cells were maintained at 37 °C in a humidified 5% CO_2_ cell culture incubator.

For compound studies, transfected HEK293T and P19 cells were exchanged into complete medium with dextran charcoal treated (DCC) bovine serum in place of normal bovine serum to deplete exogenous RA or any other non-specific hormones for 18 h before compound treatment and downstream experiments. For P19 compound experiments, the 7.5% calf serum and 2.5% fetal bovine serum (FBS) mixture was replaced with 10% DCC fetal bovine serum.

### 4.4. RAR Luciferase Reporter Assay

Luciferase assay for RAR activation was performed as described in [58]. Briefly, Cos-1 cells were transfected with RARE-tK-Luc and pRL renilla control plasmid using lipofectamine 3000 (Invitrogen). Following transfection, cells were washed and exchanged into fresh maintenance medium and were treated with DMSO control, RA, or compound at 0.25 μM for 24 h. Luciferase assay was performed using the Dual-Luciferase Reporter Assay kit (Promega, Madison, WI, USA). Luciferase and renilla signal was detected on an Infinite M1000 Pro Tecan (San Jose, CA, USA) plate reader. Assay was performed at least three independent times with three replicates each time. Fold activation was determined by luciferase activity and readings were normalized to renilla internal control readings.

### 4.5. Compound Studies and Western Blot

Preliminary studies determined that 15–60 min was the optimal window to detect compound effects on CaMKII activity for HEK293T, P19, and P19-MN experiments. For each cell line, an optimal time point within 15–60 min was identified and consistently applied across all independent experiments. DMSO, atRA, C32, or C4 were then added at 0.5–5 μM for the optimal time point determined for that particular cell line, and cells were immediately harvested for western blot analyses. All compound experiments were repeated for at least three independent times.

Western blot was performed as described in [10] with the following modification: cells were immediately lysed and harvested by adding lysis buffer (9 parts: 128 mM Tris base, 10% (*v/v*) glycerol, 4% (*w/v*) SDS, 0.1% (*w/v*) bromophenol blue, pH to 6.8 and 1 part: beta-mercaptoethanol) directly to the dish or plate containing treated cells. For primary antibodies, Anti-p-CaMKII (cat #: 127165, 1/1000) was obtained from Cell Signaling, Danvers, MA, USA, Anti-CRABP1 from Sigma (cat #:HPA017203), Anti-CRABP1 from Invitrogen (Cat #: MA3-813), anti-β-Actin (cat #: SC-47778, 1/2000) was obtained from Santa Cruz Biotechnology, Dallas, TX, USA. For secondary antibodies, anti-Rabbit-IgG (cat #: 11-035-144, 1/2000) was obtained from Jackson ImmunoResearch, Ely, UK, and anti-Mouse-IgG-HRP (cat #: GTX26789, 1/5000) was obtained from GeneTex, Irvine, CA, USA.

Cell lysates were separated on 9% (*v/v*) SDS polyacrylamide gels and transferred onto 0.45 µm PVDF membrane. The membranes were cut according to molecular weight and probed with appropriate primary and secondary antibodies. Images were acquired using the Bio-Rad ChemiDoc Imager, Hercules, CA, USA (cat #: 17001402). Image analysis was performed using BioRad Image Lab software (Ver. 6.1)of ImageJ [59].

### 4.6. P19-Derived Motor Neuron (MN) Differentiation, Compound Studies, and qPCR Gene Studies

P19 cells were suspended in P19 differentiation medium (50% neurobasal medium (Gibco, Cat# 21103), 25% alpha MEM and 25% P19 maintenance medium) containing 0.5 μM retinoic acid (RA) (Sigma-Aldrich, Cat# R2625) in a T75 flask to form embryo body (EB). The flask was set up-right in the culture incubator to promote EB formation. After two days, the EBs were collected and then resuspended in P19 differentiation medium with 0.5 μM RA and 200ng/mL mouse Shh (STEMCELL Technologies, Vancouver, BC, Canada, Cat# 78066) to form neurosphere (NS). After two days, the neurospheres were dissociated into single cells by using Accumax (Millipore, Burlington, MA, USA, Cat# A7089), then suspended in P19 differentiation medium with 0.5 μM RA and 200 ng/mL Shh. The cells were plated in the 6-well plate coated with 360 μg/mL Matrigel Basement Membrane (Thermal Fisher Scientific, Cat# A1413301, Waltham, MA, USA) to differentiate into motor neurons. Brightfield images were acquired on a Leica DM IRB inverted microscope with a 10× objective lens using an Infinity 1 camera and INFINITY ANALYZE software (Ver. 6.5).

For compound studies, on the relevant day (Day 1 or Day 3), media was exchanged in depletion medium (50% neurobasal medium (Gibco, Cat# 21103), 25% alpha MEM and 25% P19 culture medium). The P19 culture medium was supplemented with 10% DCC FBS instead of normal FBS to deplete RA or other non-specific hormones that can originate from FBS.

qPCR was performed as described in [10]. Briefly, TRIzol (Invitrogen, Carlsbad, CA, USA) was used to isolate total RNA. The Omniscript RT Kit (QIAGEN, Germantown, CA, USA) was used to synthesize cDNA. qPCR was performed with SYBR-Green master mix (Agilent, Santa Clara, CA, USA) and detected with Mx3005P (Agilent). Target genes for qPCR were: *ChaT*, *Hb9*, *Isl1, Isl2*, and *Crabp1.* qPCR experiments were performed two independent times. Primer sequences can be found in Appendix A.

### 4.7. Hybrid Motor Neuron (MN1) Cell Culture and Stable CRABP1 Over-Expression Clone Generation

Wild-type MN1 cells were cultured in complete DMEM medium (Gibco #11965) containing 4.5 g/L D-glucose, 4 mM L-glutamine, 44 mM Sodium Bicarbonate, 100 U/mL penicillin, 100 mg/mL streptomycin, and 10% heat-inactivated FBS. To generate the stable over-expression MN1 cell line, first, mouse Crabp1 cDNA was cloned into pCDH-EF1α-MCS-IRES-Puro plasmid (SBI, #CD532A-2) as previously described [11], resulting in 3XFlag-HA-tagged CRABP1 as a protein product. All plasmid DNA was purified using the PureLink HiPure Plasmid Filter Midiprep Kit (Invitrogen #K210014). For lentivirus production, 2 × 10^6^ HEK-293T cells were seeded in complete DMEM medium without antibiotics dish in a 10 cm dish overnight. 9.6 µg Crabp1-pCDH target plasmid, 7.2 µg psPAX2 packaging plasmid, 2.4 µg pMD2.G envelope plasmid were co-transfected into cells with Lipofectamine 2000 transfection reagent (Invitrogen) following the manufacturer’s protocol. Media was changed to 6 mL of fresh complete DMEM medium containing 1% BSA after 6 h. Infectious lentiviruses were harvested at 24 h and 48 h post-transfection and filtered through 0.45 µM pore cellulose acetate filters. For transduction, 1 × 10^5^ MN1 cells were seeded in complete DMEM medium in 6-well plates overnight. 2 mL of lentivirus with 8 µg/mL polybrene (Millpore TR-1003-G) were added into cells, and then cells were centrifuged at 800× *g*, 37 °C for 60 min. Lentivirus was removed, and the medium was changed after 24 h. Puromycin selection was started at 48hrs post-transfection. Cells were selected and maintained in the same MN1 medium as described above with the addition of 3 μg/mL puromycin. After puromycin selection, stable MN1 cells were collected and examined for Crabp1 expression by qPCR.

### 4.8. Ionomycin-Induced Cell Death and MTT Viability Assay

MTT reagent (Sigma Cat# M5655) was prepared by dissolving in 1XDPBS to a final concentration of 5 mg/mL and then sterile-filtered. Wild-type (WT) and the CRABP1 stable clone MN1 were then seeded into a 24-well plate at a density of 1 × 10^5^ cells/well the night before. Puromycin selection was withdrawn from CRABP1-MN1 of these experiments. DMSO (vehicle) or ionomycin (5 μM) were treated with WT and CRABP1- MN1 for 18 h. The Final volume in each well was 1ml. Then 100 μL (10% of total well volume) of MTT reagent was added to MN1 cells and incubated at 37 °C in a humidified 5% CO2 cell culture incubator for three hours to allow for formazan crystal formation. Then cell culture and MTT reagent were gently removed via suction, and the remaining formazan crystals dissolved in 600 μL of DMSO. Formazan absorbance was measured at 570 nm, and a background reading at 690 nm was acquired using an Infinite M1000 Pro Tecan plate reader. Values were exported and analyzed in Microsoft Excel. After performing background subtraction, percent cell viability was calculated by the formula: (1−–(DMSO−–Ionomycin)) × * 100. This assay was performed 5 independent times with 4–12 replicates for each condition.

For compound experiments, WT-MN1 cells were seeded in a 24-well plate as described above. MN1 cells were pre-treated with RA, C32, and C4 at 0.5–5 μM for 1.5 h prior to ionomycin exposure. After pre-treatment, ionomycin was added to a final concentration of 4 μM, overnight (18 h) to induce cell death. atRA, C32, and C4 were also present in the cell culture medium for the duration of ionomycin exposure. Immediately after the 18-h co-treatment, MN1 cells were subjected to an MTT viability assay as described above. This experiment was performed three independent times with 3–4 technical replicates for each condition.

### 4.9. Statistical Analysis

HEK293T, undifferentiated P19, and P19 MN compound experiments were analyzed using paired Student’s *t*-test (DMSO vs. RA, C32, or C4). RAR activity luciferase assay data were analyzed using paired Student’s *t*-test. MTT cell viability assay data were analyzed using paired Student’s *t*-test. Significance was defined as * *p* ≤ 0.05. “N.S.” indicates not significant. Error bars for all data are presented as mean  ±  standard deviation (SD).

## Figures and Tables

**Figure 1 ijms-24-04980-f001:**
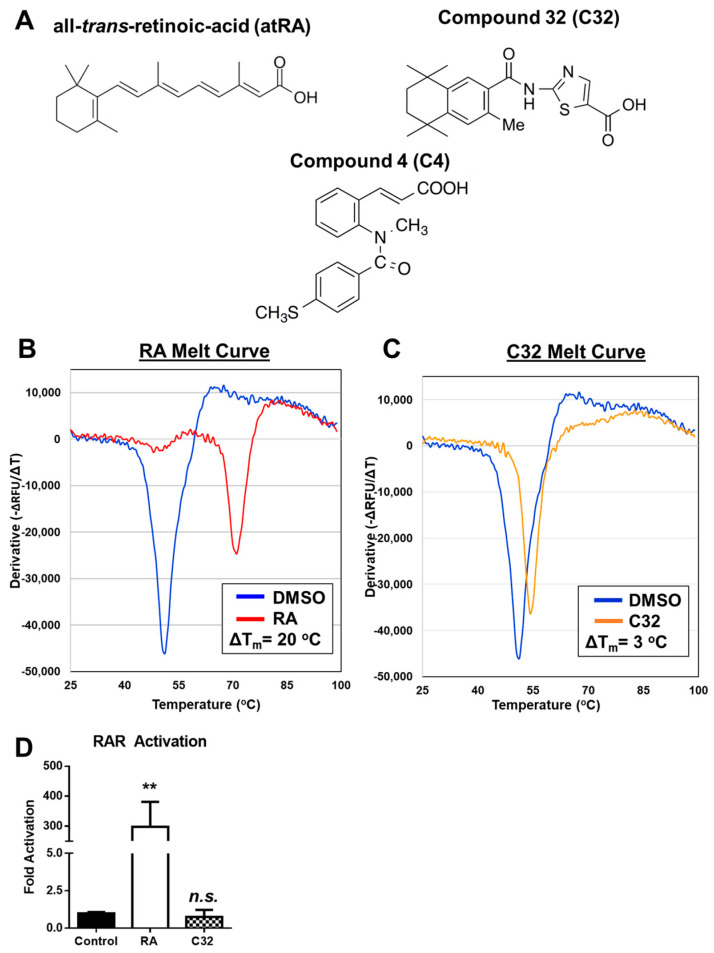
C32 is a novel CRABP1-binding compound. (**A**) Chemical structures for all-*trans*-retinoic acid (atRA), C32, and C4. (**B**,**C**) Melt curves for atRA-bound (**B**, red, 100 μM) and C32-bound CRABP1 (**C**, orange, 100 μM)) overlaid with no ligand control (blue). 5 μg of CRABP1 (14.1 μM) was used for binding assays, resulting in a ligand to CRABP1 ratio of 7:1. Ligand-induced CRABP1 melting temperature shift is reported as “ΔTm”. Melt curves are presented as the negative inverse derivative of fluorescence over temperature [−d(RFU)/dT]. Experiments were independently performed 3–4 times. (**D**) C32 does not activate RAR activity in Cos-1 cells, as determined in a luciferase reporter assay. Cos-1 cells were treated with atRA (positive control), or C32, at 0.25 μM for 24 h. A break in the y-axis shows the difference in the scale of reporter activity. *p* = 0.002 (RA) and *p* = 0.45 (C32) determined by paired Student’s *t*-test (*n* = 5). ** *p* ≤ 0.01, “*n.s.*” not significant. Error bars are presented as mean ± SD.

**Figure 2 ijms-24-04980-f002:**
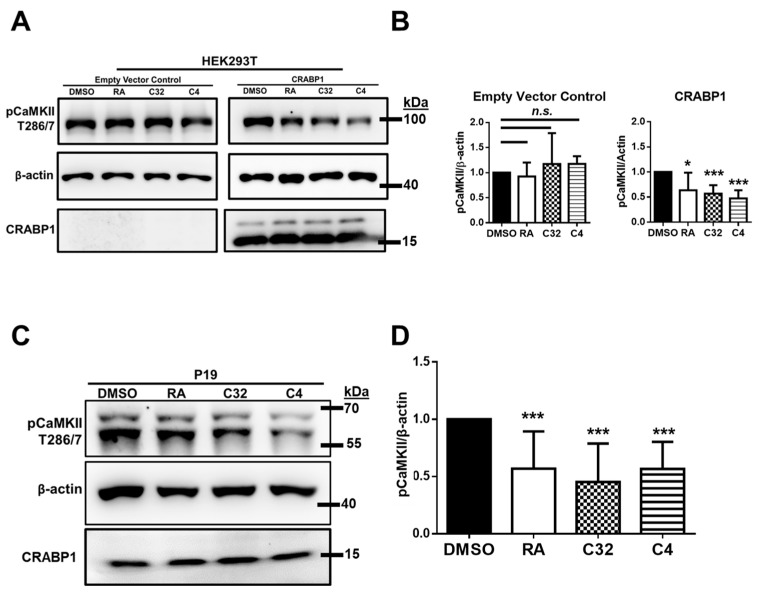
C32 and C4 dampen CaMKII activity in a CRABP1-dependent manner. (**A**,**B**) Western blot and quantification of pCaMKII activity, marked by pThr-286/7 (pCaMKII T286/7) in HEK293T cells treated with DMSO, atRA, C32, or C4 at 0.5–5 μM for 15 min. HEK293T cells were co-transfected with GFP-CaMKII and either empty vector control or CRABP1 expressing vector. Anti-CRABP1 was used to detect CRABP1 in empty vector and CRABP1 transfected samples. *p* = 0.03 (RA), 0.001 (C32), 0.001 (C4), determined by paired Student’s *t*-test (*n* = 4–7). (**C**,**D**) Western blot for detecting endogenous CaMKII activity, marked by pThr-286/7, in P19 cells treated with atRA, C32, or C4 at 0.5–5 μM for 15 min. Anti-CRABP1 was used to detect endogenous CRABP1. β-actin was used as a protein loading control. *p* < 0.0001 (atRA, C32, and C4) determined by paired Student’s *t*-test (n = 12–19). * *p* ≤ 0.05, *** *p* ≤ 0.001, “*n.s*.” not significant. Error bars are presented as mean ± SD.

**Figure 3 ijms-24-04980-f003:**
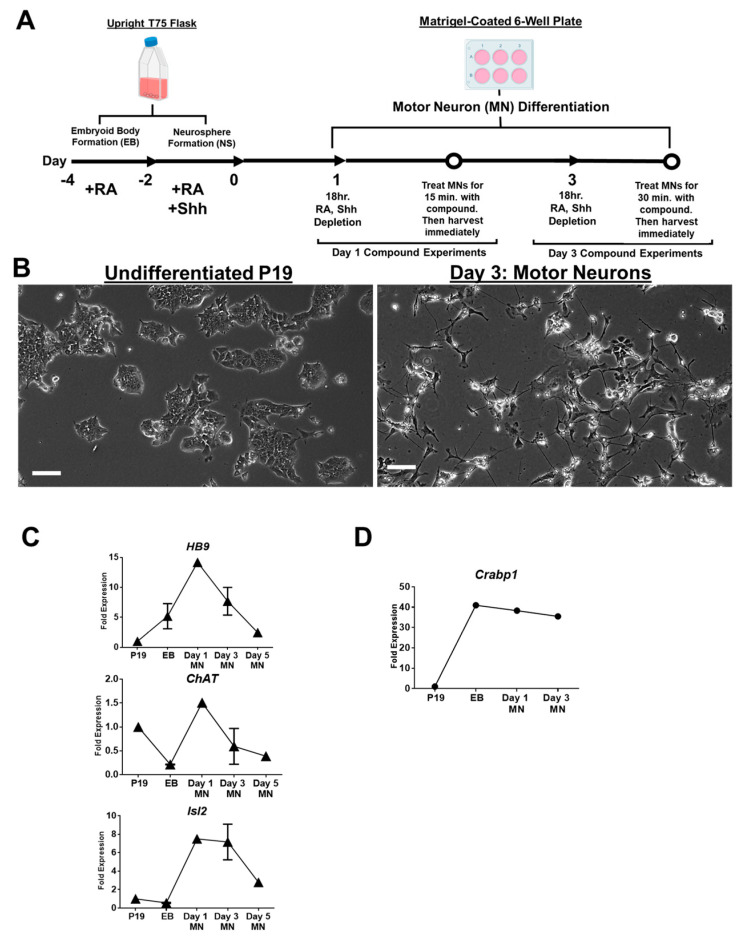
A P19-derived, in vitro MN culture system. (**A**) Workflow of P19 MN differentiation. First, adherent P19 cells were suspended in +RA (0.5 μM) medium for embryoid body (EB) formation over a two-day period. EB’s were then exchanged into +RA, +Shh medium for neurosphere (NS) formation. Optimal EB and NS formation was achieved through the use of an up-right T75 flask. NS were then dissociated and seeded onto a Matrigel-coated 6-well plate for MN differentiation. Compound experiments were conducted by first depleting atRA and Shh on the relevant day (Day 1 or Day 3) for 18 h. Immediately after the 18-h depletion (indicated by an open circle), MNs were treated with compound at 1 μM for 15 min for Day 1 MNs and 1 μM for 30 min for Day 3 MNs. MNs were then harvested for western blot analyses. (**B**) Brightfield images of undifferentiated P19 cells (left) and P19-MN differentiated cells on Day 3 (right). Scale bar (white) indicates 100 μm length. (**C**) qPCR detecting the expression of MN-specific markers, *HB9*, *ChaT*, and *Isl2*. (**D**) qPCR detecting the expression of *Crabp1*. Error bars are presented as ± SD. qPCR was conducted in two independent experiments.

**Figure 4 ijms-24-04980-f004:**
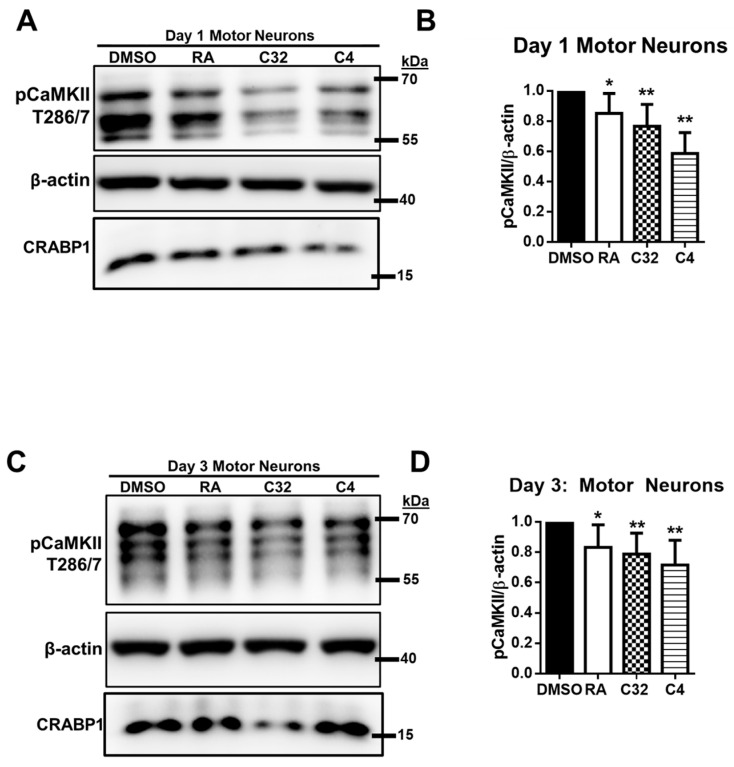
C32 and C4 dampen CaMKII activity in Day 1 and Day 3 P19-MN differentiation process. (**A**,**B**) Western blot of RA, C32, C4 effect and quantification on endogenous CaMKII activity marked by pThr 286/7 activity in Day 1 P19-differentiated MNs. Day 1 MNs were treated with 1uM atRA, C32, or C4 for 15 min and then harvested for western blot analyses. *p* = 0.02 (RA), *p* = 0.005 (C32), and *p* = 0.003 (C4) determined by paired Student’s *t*-test (n = 5–7). (**C**,**D**) Western blot and quantification of RA, C32, C4 effect on endogenous pCaMKII T286/7 activity in Day 3, P19-MNs. Day 3 MNs were treated with 1uM atRA, C32, or C4 for 30 min and then harvested for western blot analyses. Anti-CRABP1 was used to detect endogenous CRABP1. *p*= 0.04 (RA), *p* = 0.003 (C32), and *p* = 0.002 (C4) determined by paired Student’s *t*-test (n = 5–7). β-actin was used as a protein loading control. * *p* ≤ 0.05, ** *p* ≤ 0.01. Error bars are presented as mean ± SD.

**Figure 5 ijms-24-04980-f005:**
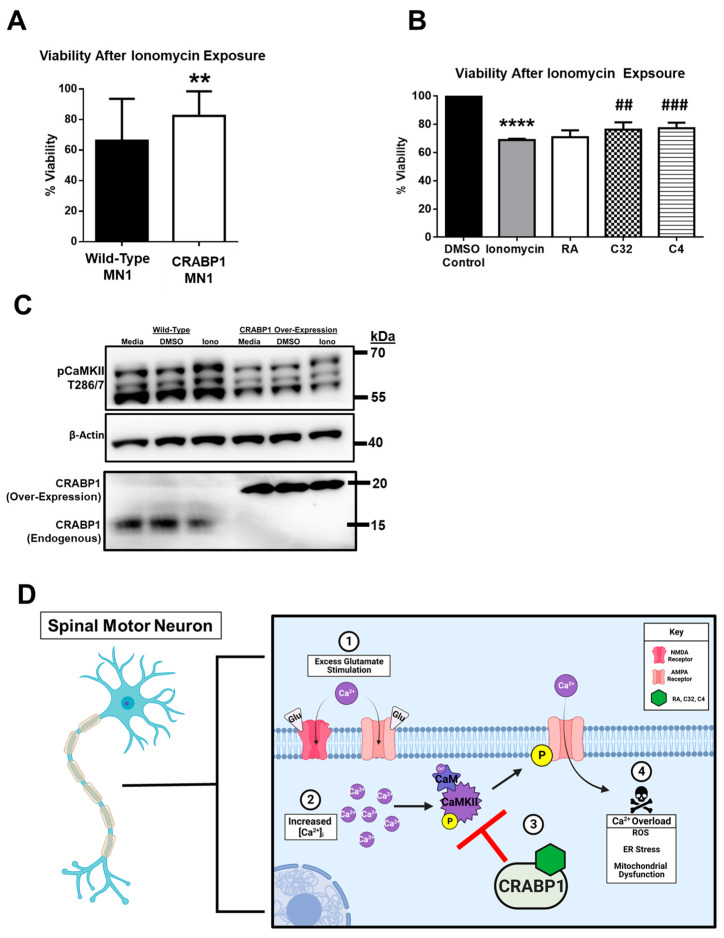
CRABP1 dampens CaMKII activity to protect against neurotoxic Ca^2+^ overload. (**A**) MTT cell viability assay of WT and CRABP1-MN1 exposed to ionomycin (5 μM, 18 h). Cell viability was measured via formazan generation with an absorbance at 570 nm. *p* = 0.01 determined by Student’s *t*-test. MTT assay was conducted in five independent experiments with 3–12 technical replicates per experiment. ** *p* < 0.01 determined by Student’s *t*-test. Values are mean ± SD. (**B**) Cell viability assay of WT MN1 to measure the protective effects of atRA, C32, and C4. WT-MN1 cells were pretreated with atRA, C32, or C4 (0.5–5 μM) for 1.5 h. Immediately after pre-treatment, ionomycin (4 μM) or DMSO (as vehicle control) was added to induce cell death and co-incubated with atRA, C32, or C4. *p* = 0.34 (RA), *p* = 0.006 (C32), *p* = 0.0004 determined with Student’s *t*-test. (C4) (**** *p* < 0.0001, DMSO vs. Ionmycin; ## *p* < 0.01 Ionomycin vs. C32; ### *p* < 0.001 Ionomycin vs. C4). Three independent experiments were performed. (**C**) Western blots of CaMKII activity marked by pThr 286/7 in wild-type (WT) and CRABP1 over-expressing MN1 cells treated with medium control, DMSO control, or ionomycin (10 μM, 5–10 min). Beta-actin was used as a loading control. Endogenous and over-expressed 3XFlag-HA-tagged CRABP1 expression was monitored with anti-CRABP1. In order to detect a much lower level of endogenous CRABP1 in WT-MN1, twice as much lysate was loaded for WT-MN1 as compared to the loaded CRABP1-MN1 (over-expression) cell lysate. (**D**) A model depicting the protective role of CRABP1 in MNs when they are exposed to (1) excitotoxicity which results in (2) pathological increases in [Ca^2+^]I (purple circles), and subsequent CaMKII over-activation and phosphorylation of Ca^2+^ permeable AMPA receptors. (3) CRABP1-RA could inhibit pCaMKII over-activation, ultimately protecting cells from AMPA-mediated Ca^2+^ overload and death (4). The model was illustrated using Biorender.com.

## Data Availability

All data is available upon reasonable request from the corresponding author.

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
