# Peer review of "Targeting Cellular Retinoic Acid Binding Protein 1 with Retinoic Acid-like Compounds to Mitigate Motor Neuron Degeneration"

_ijms, 2023, doi:10.3390/ijms24054980_

Round 1

Reviewer 1 Report

Tracking #: ijms-2195150 – Nhieu et al.

The manuscript entitled “Targeting Cellular Retinoic Acid Binding Protein 1 with retinoic acid-like compounds to mitigate motor neuron degeneration” attempts to address interesting new issues regarding the characterisation of ligands capable of binding to CRABP1 but not to the retinoic acid receptor (RAR). This study continues research published by the authors in the past.

The title clearly announces what the reader will find in the paper and the introduction allows for a good understanding of the subject.

However, authors present data from experiments that are not always well calibrated.Overall, this study may be interesting. The following is a list of points that should be clarified and/or improved before publication to make it sounder:

1/ Characterisation of compound C32 as a specific ligand for CRABP1 does not appear to be entirely conclusive.

a/ The DSF method used to evaluate ligand binding  is indeed often used in the literature. Authors write “In this assay, CRABP1 binding was
defined as a ΔTm greater than, or equal to, 1 oC (ΔTm ≥ 1 oC)”. What is the basis for this definition? Also it should be noted the very high concentration of ligand used in this experiment, 100µM.

b/ Concerning atRA, the result with the DSF method is unequivocal. For C32 the curve obtained from the measurements is very close to that of the control. Other measurement methods can be considered. First of all, since CRABP1 is purified, mass spectrometry analyses could highlight the binding of C32. Also, as the authors have already done in reference 12 to characterise the binding of C3 and C4, experiments displacing the binding of atRA could be conducted.

c/ Since it has been previously shown that C3 and C4 can displace the binding of atRA to CRABP in the reference 12, the authors should show the curves obtained in DSF with these ligands.

d/ Authors are careful to indicate that C32 does not activate RAR and do not write that C32 does not bind to RAR. However, they define C32 as a specific ligand for CRABP1. A simple experiment would be to assess the RAR antagonistic ability of C32 in the reporter gene transactivation assay by performing a competition between atRA and C32. DSF experiments could also be performed with RAR.

2/ Western blot experiments:

 a/ It is disturbing that western blots using the antibody directed against the phosphorylated form of CaMKII generate different profiles (single or multiple bands) depending on the cell types studied or the differentiation stage. Can the authors comment on these observations?

 b/ Ligand treatment times ranging from 15 to 60 minutes are reported for all these experiments. Does this indicate that not all trials were performed under the same conditions? As these are enzymatic reactions, it would be preferable to indicate the precise treatment times.

 c/ The expression of CRABP1 in P19 cells is not clearly shown but appears to be low. Figure 2C should show this expression on western blot. However, a significant effect of ligands on the phosphorylation level of CaMKII is reported. In order to attribute this effect to the binding of ligands to CRABP1, it would be desirable to perform the experiment with siRNA directed against CRABP1.

 d/ The same remarks are valid for Figure 4 concerning the P19-MN model.

3/ Additional files "Original Images for Blots/Gels" and "Download Non-published Material" are identical.

4/ At the beginning of the discussion the authors write “these results show that C32 is a CaMKII-selective CRABP1 ligand, C3 is a Erk-selective CRABP1 ligand, whereas C4 appears to be a pan-acting CRABP1 ligand”. Yet, unless I have missed the information, the effect of C32 on ERK is not addressed in this paper.

Author Response

We thank the reviewer for thorough comments. A point-to-point response has been provided in the submitted document "ijms-2195150 Response 1_Jhieu.docx"

Reviewer 2 Report

This is a nice study showing a new ligand of CRABP1, which holds great potential for motor neurons-related degenerative diseases. The conclusion about the protective role of ligand-C32 may be refined if the authors consider the following suggestions.

Specific comments:

WB, pCamkii (T286/7) showed different numbers of 1-3 bands in different cells (HEK293, P19, P19-MN). please clarify.

Fig4, Ligands treatment reduced the level of p-Camkii, and what is the effect on cell survival/differentiation. I think these changes are important.

I don't quite understand the importance of Fig5, CRABP1 overexpression suppresses p-Camkii to exert MN protection, there is no ligand-C32 treatment. And p-Camkii expression did not seem to change significantly between the Media, DMSO, and Iono groups, thus, a well-established MN-injury model to determine the effect of C32 is needed.

In addition, there was no expression of CRABP1 in WT as determined by WB, please comment.

Author Response

We thank the reviewer's thorough comments. A point-to-point response has been provided in the submitted document "ijms-2195150 Response 2_Jhieu.docx"

Reviewer 3 Report

The manuscript has been carefully prepared. I have the following comments for the authors to address.

1) The authors should highlight why their study is important and the potential clinical application of their findings

2) Has the authors performed power calculation to demonstrate they have sufficient number of replicates?

3) Please provide all raw Western blot images as supplementary materials.

Author Response

We thank the reviewer's thorough comments. A point-to-point response has been provided in the uploaded document "ijms-2195150 Response 3_Jhieu"

Reviewer #3

The manuscript has been carefully prepared. I have the following comments for the authors to address.

1) The authors should highlight why their study is important and the potential clinical application of their findings

            Response: We agree that high-lighting the clinical applications is of most importance to convey the impact of our study. We have expanded Discussion to emphasize the significance regarding clinical implication.

2) Has the authors performed power calculation to demonstrate they have sufficient number of replicates?

            Response: For all experiments, we conducted a minimum of 3 independent experiments performed on different days, unless stated otherwise. Technical replicates were also performed in experiments involving a multi-well format (e.g. MTT assay and DSF compound screening). This has been elaborated in relevant text sections. We have also provided the number of biological and/or technical replicates, as well as statistics, in figure legends.  

3) Please provide all raw Western blot images as supplementary materials.

            Response: We have provided all raw blots of the figures. Please see additional file named “Original Blots/Gels.”

Round 2

Reviewer 1 Report

I thank the authors for clearly answering the questions asked and for adding several informative data and modifying the text, which improves the paper.

Reviewer 2 Report

I have no further comments